# Euler-scale dynamical correlations
# in integrable systems with fluid motion

**Frederik S. Møller[1]⋆, Gabriele Perfetto[2], Benjamin Doyon[3] and Jörg Schmiedmayer[1]**

**1** Vienna Center for Quantum Science and Technology,
Atominstitut, TU Wien, Stadionallee 2, 1020 Vienna, Austria
**2** SISSA – International School for Advanced Studies and INFN,
via Bonomea 265, 34136 Trieste, Italy
**3** Department of Mathematics, King's College London, Strand, London WC2R 2LS, UK

⋆ frederik.moller@tuwien.ac.at

## Abstract

We devise an iterative scheme for numerically calculating dynamical two-point correlation functions in integrable many-body systems, in the Eulerian scaling limit. Expressions for these were originally derived in Ref. [1] by combining the fluctuation-dissipation principle with generalized hydrodynamics. Crucially, the scheme is able to address non-stationary, inhomogeneous situations, when motion occurs at the Euler-scale of hydrodynamics. In such situations, in interacting systems, the simple correlations due to fluid modes propagating with the flow receive subtle corrections, which we test. Using our scheme, we study the spreading of correlations in several integrable models from inhomogeneous initial states. For the classical hard-rod model we compare our results with Monte-Carlo simulations and observe excellent agreement at long time-scales, thus providing the first demonstration of validity for the expressions derived in Ref. [1]. We also observe the onset of the Euler-scale limit for the dynamical correlations.

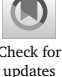

# 1   Introduction

With the advent of experimental realizations of cold gases in reduced dimensions, the study of many-body systems far from equilibrium has received a lot of attention [2–14]. Among such low-dimensional systems, the class of integrable models, admitting an infinite set of conservation laws in the thermodynamic limit, is of particular interest. In the framework of homogeneous quantum quenches [15–20], an isolated many-body system with short-range interactions on infinite volume undergoes unitary time evolution starting from a state that is homogeneous (i.e. translationally invariant) and clustering (i.e. with short-range correlations). It is by now well understood that relaxation to a stationary state occurs in the long time limit, at least under appropriate time averaging. It is expected that the space of possible stationary states be completely determined by the set of quasi-local conserved quantities. Depending on the symmetries of the Hamiltonian, two main scenarios are known. In non-integrable systems, on the one hand, where one assumes that only a finite number of quasi-local conserved quantities exist, including the Hamiltonian, chaotic motion causes relaxation to occur to a thermal state described by the Gibbs ensemble, with a finite number of thermodynamic potentials. Thermalization in this framework may be understood within the eigenstate thermalization hypothesis, first proposed in Refs. [21,22] (see Ref. [23] for a review on the subject). In integrable systems, on the other hand, relaxation in general occurs to stationary states which display non-thermal features, as a consequence of the infinite number of conservation laws. The natural stationary states of integrable systems are the so-called generalized Gibbs ensembles (GGE), as first proposed in Ref. [24] (with the excellent reviews on the subject Refs. [19,25]). In this perspective, one speaks for integrable models about "generalized thermalization" [26] since GGEs generalize the canonical Gibbs ensembles by admitting an infinite number of Lagrange parameters, associated to the infinite number of conserved quantities. Integrable models of particular interest include the Lieb-Liniger model describing one-dimensional Bose gases [27–29], the XXZ Heisenberg spin chain [30,31], the classical and quantum sinh-Gordon field theory [32–35], and many more [36–39]. The full thermodynamics of such integrable models in GGEs can be accessed via the thermodynamic Bethe ansatz (TBA) [29,32,40–42], see also Ref. [43] for a review in the context of cold atomic gases. Beyond the paradigm of infinite volume quenches from clustering states, the set of stationary states under large time averaging is much richer, such as in incompletely chaotic systems [44].

Studying the dynamics of *inhomogeneous* quantum quenches, where the initial state is not translation invariant, is much more difficult. In this perspective, a ground-breaking advance-

ment has been the introduction of the *generalized hydrodynamics* (GHD) theory in Refs. [45, 46], which allows for the analytical study of such dynamics in interacting integrable models using the concept of emergent hydrodynamics. Accordingly, for states changing significantly only on length scales much larger than the microscopic scales, the system is described as a set of fluid cells at space-time points $(x, t)$ (with $x$ ($t$) denoting the space (time) coordinate). Fluid cells are at mesoscopic scales; much greater than microscopic scales, but much smaller than macroscopic observation and variation scales. In each fluid cell, by generalized thermalization, relaxation to a local, $(x, t)$-dependent GGE is assumed; technically, each cell is described by TBA. The rationale is that within fluid cells relaxation occurs on time-scales faster than the global dynamics. At large enough times, but before thermodynamic stationarity is reached, there is a long period of large-scale motion well described by hydrodynamics.

GHD is the hydrodynamic theory based on GGEs instead of canonical Gibbs ensembles, thus correctly accounting for the infinite number of emerging hydrodynamic modes. In this paper, we are interested in its "Euler-scale", obtained in the exact Euler scaling limit, where the space-time observation points and the initial state's variation lengths are sent to infinity simultaneously [47]. GHD has already proven itself immensely successful [48], and several additions have been made to the theory allowing it to describe the spreading of entanglement [49–52], diffusive effects [53–56], correlation functions [1, 57, 58], and much more [59–67].

In many-body systems, dynamical correlation functions – measuring the correlations between observables at different space-time points – are of particular interest. They provide a powerful way of characterizing quantum and classical systems, encoding the emergent degrees of freedom and their dynamics. Unfortunately, they are difficult both to calculate theoretically from first principles, and to extract experimentally [10, 12, 68]. The frameworks of linear and nonlinear fluctuating hydrodynamics [69], and the theory of hydrodynamic projections [47, 59], provide powerful methods able to access the large-scale behaviours of dynamical correlation functions. By focusing on the Euler-scale hydrodynamic equations and the propagation of long-lived modes, they allow one to extract exact asymptotic expressions for correlation functions along the propagation of such modes. However, these well-developed methods are limited to correlation functions in homogeneous and stationary states: they are based on (non)linear response mechanisms with respect to such states. In particular, a natural question arises as to correlations within many-body systems with large-scale, hydrodynamic motion.

The theories of inhomogeneous Luttinger liquids [70–76] and, more recently, quantum GHD [63] are able to account for quantum critical correlations in such cases, which dominate at zero temperature or more generally in zero-entropy states (not necessarily of Gibbs form). At finite entropy, the dominant correlations are instead due to classical fluctuations[1], both in quantum and classical systems. For these, a combination of the fluctuation-dissipation principle and GHD lead to a recursive procedure for generating $n$-point correlation functions at the Euler-scale [1]. Crucially, the method gives results valid also when the system displays nontrivial Euler-scale motion. Given the universality of GHD, the results obtained should be applicable to any integrable model obeying the hydrodynamic equations. Through the developed technique, exact dynamical two-point correlations for a large class of local fields were derived. However, numerical checks were made only in stationary states [58, 77]. The much more nontrivial part of the theory of Ref. [1] is that dealing with correlations within many-body systems with Euler-scale motion, where correlations depend on the full initial GHD profile. The results take the form of a quite involved set of nonlinear integral equations, which were never

---

[1]Recall, though, that correlations due to classical fluctuations, that is, to the statistical distribution of states, keep a memory of quantum effects in quantum systems: the statistics of the underlying degrees of freedom affect the distribution function. Thus, even at finite entropy, in quantum systems correlations depend on $\hbar$ (set to 1 in this paper).

solved numerically, let alone verified against microscopic simulations. These equations encode both the correlations due to fluid modes propagating along the flow, as well as subtle, "indirect" corrections resulting from the nonlinearity of the fluid equations, which are present if the model is interacting.

In this paper, we develop a numerical scheme for calculating dynamical two-point correlations in Euler-scale GHD, in the full generality of the theory of Ref. [1]. We use the scheme to describe the propagation of correlations from inhomogeneous states in the Lieb-Liniger model, the relativistic sinh-Gordon model, and the classical hard-rod gas. To confirm the validity of our method and of the predicted analytical expressions, we compare our results for the hard-rod gas to Monte-Carlo simulations and examine the time required to reach the Euler-scale of correlations. All aspects of the predicted correlations are reproduced, including the subtle indirect corrections.

The paper is organized as follows. In Sec. 2, we briefly review the basics of GHD, while the correlation functions originally derived in Ref. [1] are discussed in Sec. 3. Next in Sec. 4, we calculate and analyze the spreading of correlations in four distinct systems: the Lieb-Liniger model in an homogeneous thermal state, Subsec. 4.1, a bump releases in the classical hard-rod gas, Subsec. 4.2, and in the relativistic sinh-Gordon quantum field theory, Subsec. 4.3, and the partitioning protocol in the Lieb-Liniger model, Subsec. 4.4. Finally, we conclude in Sec. 5. The technical aspects of the presentation are deferred to the Appendix.

## 2 Summary of GHD

Although the theory of generalized hydrodynamics is still relatively young, several works have provided reviews, see for example Refs. [78, 79] and the lecture notes [48]. In this Section we simply reiterate some of the central concepts of GHD, which later will be relevant for computing dynamical correlation functions. Note that various notational conventions have been used in the literature; here we will follow that of [79].

Generalized hydrodynamics describes the transport properties of integrable systems, which due to their infinite set of conservation laws exhibit dynamics very different from conventional hydrodynamics [13, 45, 46, 80, 81]. The theory employs the language of the thermodynamic Bethe ansatz (TBA), where the full thermodynamics of an integrable system is encoded in a root density, $\rho_{\mathrm{p}}(\lambda)$. The root density can be interpreted as a density of quasi-particles parametrized by the rapidity $\lambda$ [29, 32, 40, 41], and it characterizes the generalized Gibbs ensemble. The information of the Lagrange parameters associated to the conserved quantities is encoded within the so-called pseudoenergy function $\varepsilon(\lambda)$, via the non-linear integral equation

$$\varepsilon(\lambda) = w(\lambda) - \int \mathrm{d}\lambda' \, T(\lambda, \lambda') \mathrm{F}(\varepsilon(\lambda')), \tag{1}$$

where $T(\lambda, \lambda') = \partial_\lambda \Theta(\lambda, \lambda')/2\pi$ is the differential two-body scattering phase (assumed to be symmetric for simplicity), with $\Theta(\lambda, \lambda')$ the scattering phase[2]. Here, the function $\mathrm{F}(\varepsilon)$ is the "free energy function" encoding the statistics of the quasiparticles of the theory. For fermions $\mathrm{F}(\varepsilon) = -\log(1 + e^{-\varepsilon})$ (as in the case of the Lieb-Liniger model and the sinh-Gordon) and $\mathrm{F}(\varepsilon) = -e^{-\varepsilon}$ for classical particles (as for the hard-rod gas); and the source term $w(\lambda)$ is given by

$$w(\lambda) = \sum_i \beta_i \, h_i(\lambda), \tag{2}$$

where $h_i(\lambda)$ is the single particle eigenvalue of the conserved charge $Q_i$ (and thus fully specifies $Q_i$) and $\beta_i$ the corresponding Lagrange parameter. The relation between the pseudoenergy

---

[2]Note the sign difference in the definition of $T(\lambda, \lambda')$ with respect to the convention of, for instance, [48].

$\varepsilon(\lambda)$ and the quasi-particle density $\rho_{\mathrm{p}}(\lambda)$ is obtained via the filling fraction $\vartheta(\lambda) = d\mathrm{F}(\varepsilon)/d\varepsilon\big|_{\varepsilon=\varepsilon(\lambda)}$, with the nonlinear integral equations

$$\vartheta(\lambda) = \rho_{\mathrm{p}}(\lambda)/\rho_{\mathrm{s}}(\lambda) \quad , \quad 2\pi\rho_{\mathrm{s}} = (\partial_\lambda p)^{\mathrm{dr}} . \tag{3}$$

Here $p(\lambda)$ is the one-particle momentum, and the superscript $^{\mathrm{dr}}$ denotes the dressing operation, defined by the linear integral equation

$$h^{\mathrm{dr}}(\lambda) = h(\lambda) - \int d\lambda' \, T(\lambda, \lambda')\vartheta(\lambda')h^{\mathrm{dr}}(\lambda') . \tag{4}$$

The dressing encodes a modulation of the quantity induced by interactions between the quasi-particles.

The thermodynamic Bethe ansatz describes the natural stationary states of integrable systems. Used within Eulerian hydrodynamics, the root density is now seen as a dynamical function, propagating in space-time. Within fluid cells, the GGEs are described by $(x, t)$-dependent root densities, $\rho_{\mathrm{p}}(x, t; \lambda)$, or equivalently $(x, t)$-dependent filling fractions $\vartheta_t(x; \lambda)$. In the absence of any inhomogeneous external fields, the dynamics of the system is given by a simple, Eulerian fluid equation [45, 46]

$$\partial_t \vartheta_t(x; \lambda) + v^{\mathrm{eff}}(x, t; \lambda)\partial_x \vartheta_t(x; \lambda) = 0 . \tag{5}$$

The local velocity field $v^{\mathrm{eff}}(\lambda)$ in Eq. (5) has been first introduced in Ref. [82] and it specifies the propagation velocity of a test quasi-particle within the GGE bath. It takes the form

$$v^{\mathrm{eff}}(\lambda) = \frac{(\partial_\lambda \epsilon)^{\mathrm{dr}}}{(\partial_\lambda p)^{\mathrm{dr}}} , \tag{6}$$

where $\epsilon(\lambda)$ is the one-particle energy. The effective velocity encodes the Wigner delay time, which is associated with the resulting phase shifts of elastic collisions in integrable systems [37, 83]. In the absence of interactions the particles propagate with the group velocity $v^{\mathrm{gr}}(\lambda) = \partial_\lambda \epsilon/\partial_\lambda p$, and the dressing accounts for the interactions.

The full meaning of (5) can be framed in terms of the hydrodynamic principle, see e.g. Ref. [47]. According to this, the average $\langle \mathcal{O}(x, t)\rangle_{\mathrm{inh}}$ of a local observable $\mathcal{O}(x, t)$ at the space-time point $(x, t)$, in some *inhomogeneous and non stationary* state $\langle \ldots \rangle_{\mathrm{inh}}$ where variations in space and time occur only at large wavelengths and low frequencies, can be approximated with the average of the same observable over the *homogeneous and stationary* GGE depending on the space-time point $(x, t)$. In formulas,

$$\langle \mathcal{O}(x, t)\rangle_{\mathrm{inh}} \approx \langle \mathcal{O}(0, 0)\rangle(x, t), \tag{7}$$

where we are denoting with $\langle \ldots \rangle(x, t)$ the average over the GGE at the space-time point $(x, t)$. On the right hand side of Eq. (7), the observable $\mathcal{O}(x, t)$ has been shifted to the space-time point $(0, 0)$ exploiting the fact that the GGE is homogeneous and stationary. For finite scales, say $z$, of the variation lengths and times of the inhomogeneous, non-stationary state, Eq. (7) is an approximation. It becomes exact in the Euler scaling limit, in which space and time are scaled as $z\,x, z\,t$ and $z \to \infty$. In this limit, the GGE on the right-hand side of Eq. (7) depends on the finite, rescaled space-time point $(x, t)$. In the Euler scaling limit, the system, during its evolution over Euler time-scales, remains locally in a stationary state whose time evolution is given by Eq. (5). According to Eq. (7), in particular, one-point functions can be computed once their averages over GGEs are known.

The average over a GGE of the $i$'th conserved charge $\langle \mathsf{q}_i(0,0)\rangle (x,t)$ is accessible directly from the TBA, while the average with respect to the GGE of the corresponding current $\langle \mathbf{j}_i(0,0)\rangle (x,t)$ has been obtained in Refs. [45, 46] as a central result of the GHD theory:

$$\langle \mathsf{q}_i(0,0)\rangle (x,t) = \int \mathrm{d}\lambda \, \rho_\mathrm{p}(x,t;\lambda) h_i(\lambda), \tag{8}$$

$$\langle \mathbf{j}_i(0,0)\rangle (x,t) = \int \mathrm{d}\lambda \, v^{\mathrm{eff}}(x,t;\lambda)\rho_\mathrm{p}(x,t;\lambda) h_i(\lambda), \tag{9}$$

where we recall that $h_i$ is the one-particle eigenvalue of the charge $Q_i$. It is worth noting that the exact expression of the average current $\langle \mathbf{j}_i(0,0)\rangle (x,t)$ falls outside of the historically developed TBA machinery. Equation (9) has been first derived in Ref. [45] for relativistic quantum field theories, and it has been first numerically established in Ref. [46] for the XXZ spin chain. Later, Eq. (9) has been proved for various classical models and for quantum spin chains [84–90]; in particular [86, 87] present proofs from basic hydrodynamic principles, and [89, 90] present a full Bethe ansatz theory.

Lastly, Eq. (5) admits a solution by a characteristic function, $\mathcal{U}(x,t;\lambda)$, encoding the inverse trajectories of the quasi-particles [52, 79, 91]. Given the characteristic, one can directly relate the evolved state to the initial state via the relation

$$\vartheta_t(x;\lambda) = \vartheta_0(\mathcal{U}(x,t;\lambda);\lambda). \tag{10}$$

The characteristics follow the same hydrodynamical equation as the filling function

$$\partial_t \mathcal{U}(x,t;\lambda) + v^{\mathrm{eff}}(x,t;\lambda)\partial_x \mathcal{U}(x,t;\lambda) = 0, \quad \mathcal{U}(x,0;\lambda) = x, \tag{11}$$

whereby they can be efficiently computed alongside the filling function. The characteristic function also solves a system of nonlinear integral equations [91] where time enters as a fixed parameter. This forms the basis for the recursive method leading to the exact Euler-scale dynamical correlation functions in GHD [1], which we make use of below.

Note that some TBAs admit multiple root densities (that is, multiple quasi-particle species). In this case all integrals over the rapidity transform into integrals over the rapidity of each root density, where the contribution from each root density is added together.

## 3 Exact Euler-scale dynamical two-point correlations

Euler-scale hydrodynamics describes the flow of conserved charges between locally homogeneous space-time fluid cells, whose variations to neighbouring cells are sufficiently small. Similarly, an Eulerian scaling limit for correlations exists, which is defined as the large-scale limit of connected correlation functions [1, 47], probing long wavelengths. A simple definition of the Eulerian scaling limit for correlation functions, where one of the observables is conventionally placed at time zero, is

$$\left\langle \mathcal{O}(x,t)\mathcal{O}'(y,0)\right\rangle^{\mathrm{Eul}} = \lim_{z\to\infty} z\Big( \left\langle \mathcal{O}(zx,zt)\mathcal{O}'(zy,0)\right\rangle_{\mathrm{inh},z} - \left\langle \mathcal{O}(zx,zt)\right\rangle_{\mathrm{inh},z}\left\langle \mathcal{O}'(zy,0)\right\rangle_{\mathrm{inh},z} \Big), \tag{12}$$

where $z$ is the length scale of spatial variations of the inhomogeneous state, and $x$, $y$ and $t$ are the rescaled, finite, position and time variables. In fact, the precise definition of $\left\langle \mathcal{O}(x,t)\mathcal{O}'(y,0)\right\rangle^{\mathrm{Eul}}$ needs some care: in some cases, $\mathcal{O}(zx,zt)$ and $\mathcal{O}'(zy,0)$ must be replaced by nontrivial averages over space-time fluid cells, see e.g. [92]. In the hard-rod gas analysed numerically below, a simple averaging in space is sufficient, as described in Subsec. 4.2.

In Ref. [1], a recursive procedure for evaluating such correlations was obtained based on linear responses of $\vartheta_t(x;\lambda)$ to variations in the initial state. In this method, the Eulerian dynamical correlation functions are obtained from a linear response analysis, a generalization of the fluctuation-dissipation theorem. Importantly, due to the exact solution to the problem of characteristics [91], in integrable systems, this allows one to extend the method to situations with large-scale, inhomogeneous initial states. In the Eulerian scaling limit, all equal-time, space-separated connected correlation functions vanish (at finite temperatures, microscopic correlations decay exponentially fast). However, over time the propagation of quasi-particles causes quantities in separated fluid cells to become correlated in a non-trivial manner. Hence, dynamical correlations at the Euler scale can be viewed as initial delta-function correlations, which over time ballistically spread and propagate throughout the system. This intuitive picture is reciprocated in the expression for the exact two-point Euler-scale correlation function, wherein the *propagator* $\Gamma_{(y,0)\to(x,t)}(\lambda,\lambda')$ [1] encodes the ballistic propagation of local quantities between two points in space-time. This, in general, extends to inhomogeneous states the concept from hydrodynamic projections [59].

One of the main results of Ref. [1], is the derivation of an exact formula for two-point correlations of generic local observables

$$\left\langle \mathcal{O}(x,t)\mathcal{O}'(y,0) \right\rangle^{\text{Eul}} = \int d\lambda\, \rho_{\text{p}}(x,t;\lambda) f(x,t;\lambda) V^{\mathcal{O}}(x,t;\lambda) \Big[ \Gamma_{(y,0)\to(x,t)} V^{\mathcal{O}'}(y,0) \Big](\lambda)\,. \tag{13}$$

Here, $f(\lambda)$ is the statistical factor of the model. For models with fermionic quasi-particle statistics $f = 1 - \vartheta(\lambda)$ (the case of the Lieb-Liniger and of the sinh-Gordon models), while for classical particle models $f = 1$. Meanwhile, the square brackets in Eq. (13) denote the contraction

$$\Big[ \Gamma_{(y,0)\to(x,t)} h \Big](\lambda) = \int d\lambda'\, \Gamma_{(y,0)\to(x,t)}(\lambda,\lambda') h(\lambda')\,. \tag{14}$$

Lastly, the field $V^{\mathcal{O}}$ in Eq. (13) is the one-particle-hole form factor of the operator $\mathcal{O}$. It is a functional of the filling function such that

$$-\frac{\partial}{\partial \beta_i}\langle \mathcal{O} \rangle = \int d\lambda\, \rho_{\text{p}}(\lambda) f(\lambda) V^{\mathcal{O}}(\lambda) h_i^{\text{dr}}(\lambda)\,. \tag{15}$$

The form factors must be worked out for every operator individually. For charge densities and associated currents they are very simple [59]

$$V^{\mathsf{q}_i} = h_i^{\text{dr}} \quad \text{and} \quad V^{\mathsf{j}_i} = v^{\text{eff}} h_i^{\text{dr}}\,, \tag{16}$$

however, other observables such as vertex operators in the sinh-Gordon model have form factors with more complicated expressions (see Refs. [1, 93] and Appendix B.2).

The propagator, $\Gamma_{(y,0)\to(x,t)}(\lambda,\lambda')$, describes how the local quantity $V^{\mathcal{O}'}(y,0;\lambda')$ travels through the system on a given trajectory, until it reaches the location $x$ at time $t$. The propagator itself can be split into two terms

$$\Gamma_{(y,0)\to(x,t)}(\lambda,\lambda') = \delta(y - \mathcal{U}(x,t;\lambda))\, \delta(\lambda - \lambda') + \Delta_{(y,0)\to(x,t)}(\lambda,\lambda')\,, \tag{17}$$

where each term has a clear physical interpretation. The first term denotes the *direct* propagation of the quantity carried by the quasi-particles with the inverse trajectory $\mathcal{U}(x,t;\lambda)$. Thus, of all the quasi-particles found at $(x,t)$, only those with the right rapidity have arrived from the point $(y,0)$. Meanwhile, the second term is dubbed the *indirect* propagator, as it describes modifications to the correlations due to perturbations of the quasi-particle trajectories from local inhomogeneities at $(y,0)$. Hence, all rapidities can in principle contribute to the indirect

correlations. The indirect propagator encodes subtle effects, which are due to the presence of interactions and which come from the nonlinearity of the fluid equations. One of the goals of this paper is to confirm that these effects are present and correctly described by the indirect propagator.

Inserting Eq. (17) into the two-point correlation formula of Eq. (13) yields

$$
\langle \mathcal{O}(x,t)\mathcal{O}'(y,0)\rangle^{\mathrm{Eul}} = \sum_{\gamma \in \lambda_\star(x,t;y)} \frac{\rho_s(x,t;\gamma)\vartheta_0(y;\gamma)f(y,0;\gamma)}{|\partial_\lambda \mathcal{U}(x,t;\gamma)|} V^{\mathcal{O}}(x,t;\gamma)V^{\mathcal{O}'}(y,0;\gamma)+
$$
$$
+ \int d\lambda\, \rho_{\mathrm{p}}(x,t;\lambda)f(x,t;\lambda)V^{\mathcal{O}}(x,t;\lambda)\Big[\Delta_{(y,0)\to(x,t)}V^{\mathcal{O}'}\Big](x,t;\lambda), \quad (18)
$$

where the set $\lambda_\star(x,t;y) = \{\lambda : \mathcal{U}(x,t;\lambda) = y\}$ contains only the rapidities of quasi-particles directly propagating the correlations. While the direct correlations, given by the term on the first line of the r.h.s. of Eq. (18), are relatively simple to evaluate, the indirect correlations, given by the second line of the r.h.s. of Eq. (18), pose more of a challenge. The indirect propagator follows the linear integral equation

$$
\Big[\Delta_{(y,0)\to(x,t)}V^{\mathcal{O}'}\Big](x,t;\lambda) = 2\pi \mathcal{D}_0(\mathcal{U}(x,t;\lambda);\lambda)\bigg(\Big[W_{(y,0)\to(x,t)}V^{\mathcal{O}'}\Big](\lambda)+
$$
$$
+ \int_{x_0}^{x} dz\, \Big(\rho_s(z,t)f(z,t)\Big[\Delta_{(y,0)\to(z,t)}V^{\mathcal{O}'}\Big]\Big)^{*\mathrm{dr}}(z,t;\lambda)\bigg), \quad (19)
$$

where the field $\mathcal{D}_0$ encodes the degree of inhomogeneity of the initial state (it is the "effective acceleration" first introduced in Ref. [78])

$$
\mathcal{D}_0(x;\lambda) = \frac{\partial_x \vartheta_0(x;\lambda)}{2\pi \rho_{\mathrm{p}}(x,0;\lambda)f(x,0;\lambda)}, \quad (20)
$$

and the so-called source term reads

$$
\Big[W_{(y,0)\to(x,t)}V^{\mathcal{O}'}\Big](\lambda) = -\int_{x_0}^{x} dz \sum_{\gamma \in \lambda_\star(z,t;y)} \frac{\rho_s(z,t;\gamma)\vartheta_0(y;\gamma)f(y,0;\gamma)}{|\partial_\lambda \mathcal{U}(z,t;\gamma)|} T^{\mathrm{dr}}(z,t;\lambda,\gamma)V^{\mathcal{O}'}(\gamma)
$$
$$
- \Theta(\mathcal{U}(x,t;\lambda)-y)\Big(\rho_s(y,0)f(y,0)V^{\mathcal{O}'}\Big)^{*\mathrm{dr}}(y,0;\lambda). \quad (21)
$$

Note, if the state is homogeneous $\mathcal{D}_0$ vanishes, thus eliminating any indirect correlations. In the equations above $h^{*\mathrm{dr}}(\lambda) = h^{\mathrm{dr}}(\lambda) - h(\lambda)$ and $\Theta(x)$ is the Heaviside function, while $x_0$ is an *asymptotically stationary point*, which must be chosen such that $\vartheta_s(x;\lambda) = \vartheta_0(x;\lambda)$ for $x < x_0$ and $s \in [0,t]$ [91]. Thus, $x_0$ denotes the boundary for which disturbances of correlations have yet to spread within the time $t$. One could think of $x_0 = -\infty$, although for numerical simulations it is set as the first spatial gridpoint, which must be chosen sufficiently far away from the point $y$ or any inhomogeneities.

Solving Eq. (18) requires mostly quantities already available from the TBA and GHD frameworks, however, currently no numerical solution of Eq. (19) and (21) have been shown. In Appendix A we report the technical details regarding the iterative scheme we have developed for calculating the indirect propagator in Eqs. (19)-(21) and thereby two-point correlation functions in Eq. (18). The application of this numerical scheme to several non-equilibrium protocols is presented in the next Section.

Finally, one should note that in Euler-scale GHD, one does not fully specify the observables whose correlations are evaluated: only partial information is given. For instance, conserved densities are ambiguous. Indeed, the GGE density matrix enables the exact calculation of thermodynamic averages, but contains information only of the conserved charges, $Q_i$ [24].

Thus, the conserved charge densities, $q_i$, from which the Euler-scale two-point correlation functions are derived, are defined only up to a total spatial derivative. However, in the Eulerian scaling limit any derivative corrections to $q_i$ are expected to be vanishing small, since the large-scale limit only probes long wavelengths [1].

# 4 Numerical calculation of correlations

In this Section we calculate dynamical, Euler-scale two-point correlation functions by numerically solving Eq. (18). To demonstrate properties of correlations at the Euler-scale we examine three different scenarios, whose hydrodynamical properties have already been well studied: First, we calculate the spreading of correlations from an homogeneous thermal state. The absence of inhomogeneities drastically simplifies the problem, as the indirect contribution to the correlations vanish. Next, we study how correlations spread during a bump release protocol in the classical hard-rod gas. As this is a classical model, we can simulate it via Monte-Carlo methods and microscopically measure the spreading of correlation functions, thus giving us the opportunity to test the equations for Euler-scale correlations. Additionally, we also compare the spreading of correlations during a bump release in the Lieb-Liniger model with the relativistic sinh-Gordon quantum field theory, thus highlighting the importance of the underlying dispersion relation of the model. Lastly, we examine the iconic partitioning protocol, where two homogeneous, semi-infinite systems initially at different temperatures are joined together at $t = 0$. Although partial analytic predictions for the correlations in such a setup were made in Ref. [1], these in fact contain inaccuracies. Our numerical analysis unveils the full dynamics even at short time-scales.

The exact numerical procedure for solving Eqs. (19) and (21) is fully detailed in Appendix A. Aside from the propagator, $\Gamma_{(y,0)\to(x,t)}(\lambda, \lambda')$, all other quantities of Eq. (18) are readily available via iFluid, an open-source framework for GHD numerical calculations [79]. As part of this work, the code for calculating the propagator and the two-point correlations have been integrated as a standalone module in the framework [94].

## 4.1 Homogeneous state

The spreading of Euler-scale correlations in a homogeneous system, $\vartheta_t(x;\lambda) = \vartheta(\lambda)$, is particularly simple as $\mathcal{D}_0 = 0$, causing the indirect propagator (19) to vanish. Furthermore, the velocity of the quasi-particles is spatially independent, whereby the characteristic solution to Eq. (5) becomes $\mathcal{U} = x - v^{\text{eff}}(\lambda)t$. Therefore, the the full propagator (17) reduces to

$$\Gamma_{(y,0)\to(x,t)}(\lambda, \lambda') = \delta(x - y - v^{\text{eff}}(\lambda)t)\,\delta(\lambda - \lambda')\,, \tag{22}$$

and the dynamic two-point correlation function of the zeroth charge density, $q_0 = n$, for $y = 0$ becomes

$$\langle n(x,t)n(0,0)\rangle^{\text{Eul}} = \int d\lambda\, \delta(x - v^{\text{eff}}(\lambda)t)\rho_{\text{p}}(\lambda)f(\lambda)h_0^{\text{dr}}(\lambda)h_0^{\text{dr}}(\lambda)$$

$$= t^{-1} \sum_{\lambda \in \lambda_\star(\xi)} \frac{\rho_{\text{p}}(\lambda)f(\lambda)}{|\partial_\lambda v^{\text{eff}}(\lambda)|}h_0^{\text{dr}}(\lambda)h_0^{\text{dr}}(\lambda)\,. \tag{23}$$

In Eq. (23), $\lambda_\star(\xi)$ is the set of solutions to the equation $v^{\text{eff}}(\lambda) = \xi = x/t$. Thus, the correlations spread at the same velocity as the quasi-particles move, while they diminish over time as $t^{-1}$. This formula was obtained in [59], and follows from a direct application of hydrodynamic projection methods. The algebraic decay in $t^{-1}$ is a consequence of the continuum of hydrodynamic modes (parametrised by $\lambda$) on which projection occurs – and thus this is a special

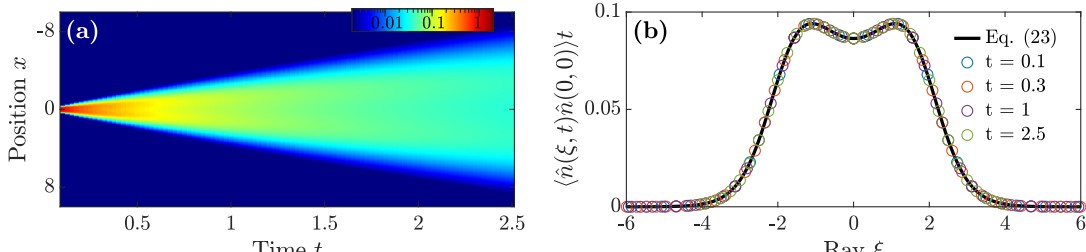

Figure 1: Two-point density correlation function of a homogeneous state in the Lieb-Liniger model. **(a)** Evolution of $\langle n(x,t)n(0,0)\rangle^{\text{Eul}}$ evaluated via Eq. (18) and plotted using a logarithmic color axis. **(b)** Time-scaled correlations at selected times $t$ plotted as circles against the ray $\xi$. The direct evaluation of Eq. (23) is plotted as a solid line and shows excellent agreement with the numerical implementation of the full formula from the homogeneous state. Simulation parameters can be found in the main text.

property found in integrable models. Note, in models like the Lieb-Liniger and sinh-Gordon model, $v^{\text{eff}}(\lambda)$ is a monotonically increasing function of $\lambda$. Hence, for any combination $(x,t)$ the set $\lambda_\star(\xi)$ will contain only one element. Computing dynamical two-point correlation functions via Eq. (23) is remarkably straightforward, as the expression can be evaluated using only information available from the TBA without performing any hydrodynamical evolution of the system.

In Fig. 1, density-density correlations calculated via the full formula (18) and simplified formula (23) are compared. The simulation was carried out for the Lieb-Liniger model with inverse temperature $\beta = 1$, interaction strength $c = 1$, and chemical potential tuned to a linear density of $\langle n(x,t)\rangle = 0.5$. Fig. 1(a) depicts $\langle n(x,t)n(0,0)\rangle^{\text{Eul}}$ and illustrates the aforementioned interpretation of the Euler-scale dynamic correlations; an initial delta function which spreads ballistically throughout the system. Since the quasi-particles move with the same velocity regardless of position and time, the solutions of the hydrodynamic equation (5) are constant on the ray $\xi = x/t$, even at short timescales. This is exemplified in Fig. 1(b), where the two-point correlation function scaled by the time, $t$, is plotted as function of the ray, $\xi$. Here, correlations calculated via the full and the simplified formula overlap perfectly, as one would expect. The shape of the two-point correlation profile features a dip towards the center originating from the statistical factor $f(\lambda)$. In the TBA of the repulsive Lieb-Liniger model, the quasi-particles are fermions (despite the model describing a Bose gas). Hence, the statistical factor reads $f(\lambda) = 1 - \vartheta(\lambda)$ and the filling function is capped at one. For sufficiently cold temperatures, a Fermi sea of quasi-particles can form at low rapidities and create a barrier for other quasi-particles trying to pass through. While the system studied here is not cold enough to form a full Fermi sea, its filling function at low rapidities is still somewhat close to unit. Therefore, the propagation of correlations at lower rapidities (and by extension low values of $\xi$) is limited causing the dip visible in Fig. 1(b).

## 4.2 Bump release and Monte-Carlo comparison

Next, we study the spreading of correlations in the classical hard-rod model. This model describes classical rods of length $a$ that propagate freely except for elastic collisions, at which rods exchange their velocities. The TBA functions describe the velocity tracers, the tracers of rods at a given velocity $\lambda$. These propagate with linear trajectories interrupted by actual jumps occurring when collisions happen. The main ingredients of the TBA description of this model, first developed in Refs. [36,59], are reported in the Appendix B.3. Fundamentally, thanks to its

classical properties, we can directly measure the spreading of connected correlation functions via classical Monte-Carlo simulations and therefore compare with the Euler-scale formulas.

For this demonstration, we turn to another well-studied protocol, namely the release of a density bump, initially located around $x = 0$, created by an inhomogeneous temperature profile. In addition, we also consider the more intricate case of the release of two bumps that initially do not overlap on top of a thermal background. Both setups are akin to what was studied experimentally in Ref. [13]. In the Appendix C further results for the release of two density bumps are presented. The initial state is a thermal GGE identified by the source term $w^{(th)}(x, \lambda) = \beta(x)\lambda^2/2$, cf. Eqs. (1), (2) and $h(\lambda) = \lambda^2/2$ the single particle energy eigenvalue for a Galilean model in Eq. (4), with $\beta(x)$ for the two bumps problem

$$\beta(x) = \beta_{as} + (\beta_{in} - \beta_{as})e^{-((x-x_0)/z)^2}\Theta(x) + (\beta_{in} - \beta_{as})e^{-((x+x_0)/z)^2}\Theta(-x), \qquad (24)$$

while for the single bump case a single Gaussian profile is considered

$$\beta(x) = \beta_{as} + (\beta_{in} - \beta_{as})e^{-(x/z)^2}. \qquad (25)$$

$z$ is the length scale where the initial rods' distribution varies as a function of space according to the definition in Eq. (12) and it controls the smoothness of the bump space dependence, $\pm x_0$ are the bumps initial positions (for simplicity we take them symmetric with respect to the origin) and $\beta_{as}$, $\beta_{in}$ are the thermal background and the bump inverse temperatures, respectively. The thermal root density $\rho_p^{(th)}$ of the initial state at time $t = 0$ reads

$$\rho_p^{(th)}(x, 0, \lambda) = \frac{\exp[-w^{(th)}(x, \lambda) - W(a\,d(\beta(x)))]}{2\pi[1 + W(a\,d(\beta(x)))]}, \qquad (26)$$

where $d(\beta) = 1/\sqrt{2\pi\beta}$, whereby the initial linear density of particles $n(x, 0)$ is

$$n(x, 0) = \int_{-\infty}^{\infty} d\lambda' \rho_p^{(th)}(x, 0, \lambda') = \frac{W(ad(\beta(x)))}{a[1 + W(ad(\beta(x)))]}. \qquad (27)$$

Here $W(z)$ is the Lambert W function on its principal branch [95]. In the Monte-Carlo simulations, rods are at the initial time $t = 0$ distributed in space according to Eq. (27) starting from some initial point $-L$ ($L > 0$), while the velocity of each rod is drawn from a Gaussian distribution with variance $1/\beta(x)$ dependent on the point $x$ where the rod is initially located, according to Eqs. (24) (or (25)) and (26). From this initial condition, we then run the deterministic classical dynamics of the hard-rod gas. For each sample of the initial condition, the linear particle density $\langle n(x, t)\rangle$ ($h(\lambda) = 1$ in Eq. (4)), and the density-density connected correlation function $t\langle n(x, t)n(0, 0)\rangle^c$ multiplied by time $t$ are acquired by counting for each space point $x$ the number of particles in an interval $(x - l/2, x + l/2)$ of length $l$ both at time 0 and after some time $t$. The average of the aforementioned quantities with respect to many independent realizations of the initial rods' positions and velocities is eventually computed. We stress that only the initial configuration of the particles is random, while the dynamics are completely deterministic. More details about the Monte-Carlo simulations are in Appendix C.

The parameter $z$ has to be chosen big enough such that $\rho_p^{(th)}(x, 0, \lambda)$ is smooth and a sufficiently large number of rods are contained within the bump. In this way one can then expect that the root density $\rho_p^{(th)}(x, 0, \lambda)$ in Eq. (26) can be propagated in time according to the GHD equations (5). The bumps initial positions $x_0$ are consequently to be taken large so that the two bumps do not overlap. The bump inverse temperature $\beta_{in}$ is fixed so that the density close to the bumps $x \sim \pm x_0$ is high and rods are densely distributed, thereby making interactions among the particles important for the dynamics. The thermal background density is set by $\beta_{as}$

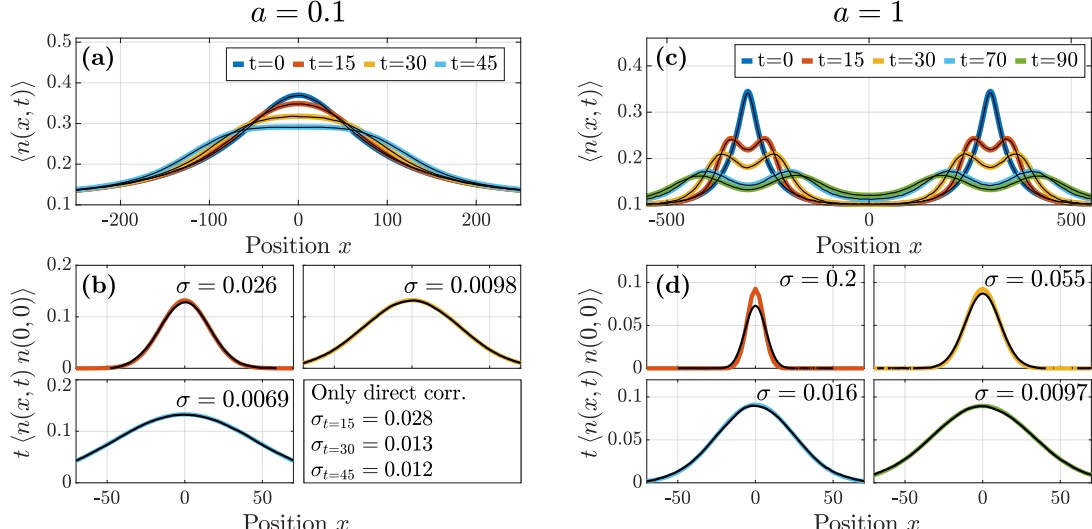

Figure 2: Bump releases in the classical hard-rod model for two different rod lengths $a = 0.1$ ($a = 1$). Results were calculated using GHD (colored lines) and Monte-Carlo methods (black lines). Parameters of the Monte-Carlo simulations are specified in the main text. **(a,c)** Comparison of the evolution of the linear density. **(b,d)** Comparison of the two-point correlation function $t \langle n(x,t)n(0,0) \rangle$ along with the distance between the two methods, $\sigma$ in Eq. (28). The additional panel shows the distances when only accounting for the direct correlations (see Eqs. (17) and (18)).

and is needed to avoid to consider space regions with no particles, which could cause non-Eulerian effects. In particular, $\beta_{as}$ is taken larger than $\beta_{in}$ in order for the background density to be smaller than the bump density. Furthermore, since the variance of the rods velocity distribution is $1/\beta(x)$, particles from the background density intervals move slower than the ones initially located in the bumps and the dynamics is therefore characterized by the propagation of the particles from the hot high-density bump regions to the cold low-density background. In particular, for short times each of the two density peaks evolves independently of the other, while for large enough times the density in the central background region, around $x = 0$, increases as a consequence of the arrival of the rods from both the bumps, thereby inducing correlations among the particles coming from the left and the right density peak.

For the single-bump the parameters used in the Monte-Carlo simulations are the following: $N = 200$, $z = 200$, $\beta_{in} = 1$, $\beta_{as} = 10$, $a = 0.1$, $L = 460$ and $l = 10$. The number of samples $M$ is $1.5 \cdot 10^6$ for $t = 15$, $5 \cdot 10^6$ for $t = 30$ and $12 \cdot 10^6$ for $t = 45$. Meanwhile, the parameters of the double bump release read: $x_0 = 300$, $\beta_{as} = 10$, $z = 120$, $\beta_{in} = 0.2$ and $a = 1$. The number of rods used in the Monte-Carlo simulations is $N = 210$, $L = 660$ and $l = 10$. For $t = 15$ and $t = 30$ we use $2 \cdot 10^6$ samples, while for $t = 70$ and $90$, since the noise in the simulations increases, the sampling is enlarged to $7 \cdot 10^6$ and $8 \cdot 10^6$ samples, respectively.

The results are shown in Fig. 2, where they are compared against GHD predictions for various values of time $t$. One can see in Fig. 2 that the time evolution $\langle n(x,t) \rangle$ of the density from the initial condition (24) ((25)) with (27) matches the GHD predictions for all the times $t$ values displayed in the figure.

Meanwhile, for the two-point correlations at short times ($t = 15$ in Fig. 2(b) and $t = 15, 30$ in Fig. 2(d)), discrepancies between the Monte-Carlo simulations and the Euler-scale results are evident. These deviations are expected to be related to diffusive corrections to the Euler-scale results in Eqs. (18)-(21), which are relevant at small time-scales [47]. The Euler-scale

two-point correlation functions of Eqs. (18)-(21) must be supplemented, at short time-scales, by the corresponding diffusive corrections in order to correctly reproduce the results of the Monte-Carlo simulations. Nevertheless, the discrepancies observed between the Euler-scale predictions and Monte-Carlo results are still relatively small, even at short time-scales – diffusive corrections are small.

In order to analyse the approach to the predicted Eulerian correlation function, we quantify this discrepancy by looking at the relative distance $\sigma$ between the results of the two methods (with $\langle n(x,t)n(0,0)\rangle^c_{\mathrm{MC}}$ defined in Eq. (52) of Appendix C)

$$\sigma = \frac{\left[\int \mathrm{d}x \ \left(t \langle n(x,t)n(0,0)\rangle^{\mathrm{Eul}} - t \langle n(x,t)n(0,0)\rangle^c_{\mathrm{MC}}\right)^2\right]^{1/2}}{\left[\int \mathrm{d}x \ \left(t \langle n(x,t)n(0,0)\rangle^{\mathrm{Eul}}\right)^2\right]^{1/2}}, \tag{28}$$

which is reported in Fig. 2(b),(d). In particular, the small value of $\sigma$, in Fig. 2(b) for the curve at $t = 15$ and in Fig. (d) for the curves at $t = 15, 30$, is mostly caused by the discrepancy between the Euler-scale predictions and the Monte-Carlo simulations around the space point $x = 0$. These differences are, on the contrary, absent for longer time-scales ($t = 30, 45$ in Fig. 2(b) and $t = 70, 90$ Fig. 2(d)) so that correlations are well reproduced by their Euler-scale limit in Eqs. (18)-(21). This is expected, as the distribution of rods grows smoother over time, whereby the Euler-scale description becomes increasingly accurate in accordance to the discussion done after Eq. (7). Additional comments regarding how the deviations from the hydrodynamic regime depend on the smoothness of the distribution (or generally, the length scale $z$) can be found in Appendix C.

As mentioned, a subtle aspect of Eq. (18) is the presence of the indirect propagator (19). The direct propagator, the first term in Eq. (18), represents the direct contribution of the normal modes (the quasi-particles), where correlations are due to the direct transport of quasi-particle along their curved trajectories within the inhomogeneous, non-stationary state. The indirect propagator is a correction to this, and it is due to the nonlinearity of the GHD equations: in a linear-response picture of the correlation function, it encodes the effects of the local disturbance of normal mode $\lambda$ on normal mode $\lambda'$. In our numerical analysis, we observe that this correction is extremely small, the dominant part of the correlation function coming from direct propagation. However, the correction is nonzero, and, as we report in Fig. 2(b), neglecting it renders the agreement with the simulation slightly worse. The subtle effect of indirect propagation is therefore explicitly observed.

We stress that this is the first comparison against numerical simulations of the formulas for the inhomogeneous Euler-scale correlation functions in Eqs. (18), (19) and (21) of Sec. 3. In the simpler homogeneous thermal framework, Euler-scale correlation functions have been compared in Ref. [92] against Monte-Carlo simulations for the classical sinh-Gordon field theory. In the latter case, results of the simulations oscillate at all times around the GHD predictions and fluid cell averaging is necessary in order to integrate them out. In the present study, on the contrary, the agreement between the classical simulations of the dynamics and the hydrodynamic expression of correlation functions become evident at larger times without the need of any further averaging procedure.

## 4.3 Comparing light cones of different models

As observed in the previous Sections, the two-point Euler-scale correlations are dominated by their direct contributions arising from the propagation of the quasi-particles. Therefore, the spreading of correlations within a given model is highly dependent on its underlying dispersion relation. This is illustrated perhaps most clearly by comparing the spreading of correlations in a relativistic and in a non-relativistic model prepared in similar initial states.

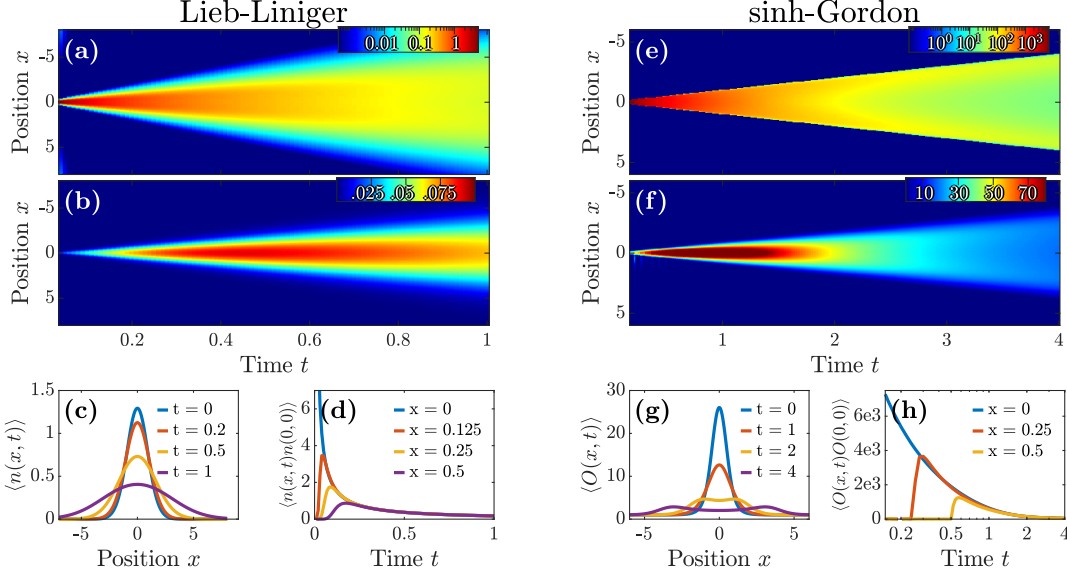

Figure 3: Two-point correlations, $\left\langle \mathcal{O}(x,t)\mathcal{O}'(0,0) \right\rangle^{\mathrm{Eul}}_{[\vartheta_0]}$, of bump releases in the Lieb-Lininger and sinh-Gordon model. The chosen operators of the Lieb-Liniger model are $\mathcal{O} = \mathcal{O}' = n$, while they for the sinh-Gordon are $\mathcal{O} = \mathcal{O}' = \Phi_2$. **(a,e)** *Direct*, dynamical two-point correlations plotted on logarithmic color axis. **(b,f)** *Indirect*, dynamical two-point correlations. **(c,g)** Evolution of the operator expectation value. **(d,h)** Evolution of the full correlations at various points in space.

In this Section, specifically, we examine a bump release in the relativistic sinh-Gordon quantum field theory [96] and in the non-relativistic Lieb-Liniger model. Both bumps were realized using inhomogeneous chemical potentials: for the Lieb-Liniger model, the inverse temperature is $\beta = 0.25$, the coupling is $c = 1$, and the chemical potential $\mu(x) = 2 - 2x^2$. For the sinh-Gordon model the inverse temperature is $\beta = 0.25$, and we have $\alpha = 0.0369$, $m = 0.9989$, and $\mu(x) = 2 - 2x^2$. The TBA of the sinh-Gordon model is reported in Appendix B.2. Additionally, Eq. (13) describes the two-point correlation functions between any two operators, as long as their one-particle-hole form factor is known. Therefore, in the sinh-Gordon model, we calculate correlations between the vertex operators $\Phi_k(x,t) = e^{kg\phi(x,t)}$, where $k \in \mathbb{R}$, while $g$ is the interaction parameter and $\phi(x,t)$ the quantum field of the model (see Appendix B.2). In this case we set $k = 2$.

Fig. 3 displays the resulting correlations from the aforementioned bump release, and immediately we observe very different spreading of direct correlations in the two models. In the Lieb-Liniger model, the quasi-particle group velocity is directly proportional to its rapidity. Thus, there is no upper bound to the velocity, and the emerging light cone of the direct correlations has a smooth edge. Meanwhile, the group velocity in the relativistic sinh-Gordon model is bounded, as it scales a $v^{\mathrm{gr}}(\lambda) \sim \tanh \lambda$. Thus, the light cone of its direct correlations has a characteristic sharp edge. Interestingly, the indirect correlations of the two models are fairly similar, and do not reflect the quasi-particle velocities to the same extend. Instead, the indirect correlations are mainly determined by the inhomogeneity of the system, which is fairly similar in the two cases (both are bump releases). Unlike the direct correlations, the indirect correlations do not decrease monotonically but in fact increase at first. We can understand this from the definition of the indirect correlations, namely how they are a consequence of the change in quasi-particle trajectories due to inhomogeneity. Over time, more and more particle will

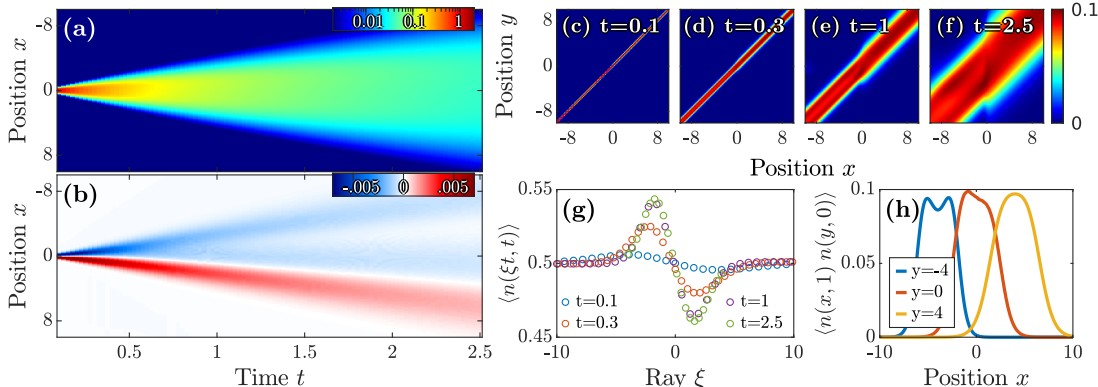

Figure 4: Two-point density correlation function of partitioning protocol in the Lieb-Liniger model. **(a)** *Direct*, dynamical two-point correlations for $y = 0$ plotted on logarithmic color axis. **(b)** *Indirect*, dynamical two-point correlations for $y = 0$. **(c-f)** Time-scaled correlation matrices, $t \langle n(x,t)n(y,0) \rangle^{\text{Eul}}$, at selected times. **(g)** Linear density as function of the ray $\xi = x/t$ at selected times, $t$. **(h)** Cut-outs of the correlation matrix at $t = 1$ for different values of $y$. Simulation parameters can be found in the main text.

cross the point $y = 0$, thus increasing the indirect correlations. Meanwhile, as the correlations disperse, they start trailing of as $\sim t^{-1}$. These two competing effect produces the light cones observed in Fig. 3.

## 4.4 Partitioning protocol

Finally, we turn our attention to the well-known partitioning protocol [45, 46], where two homogeneous, semi-infinite systems are stitched together at the space-time point $(x = 0, t = 0)$. The two subsystems have different initial root densities causing a flow of charges between the two subsystems once they are joined together. In this example we study a partitioning protocol in the Lieb-Liniger model, where two subsystems of different temperature $\beta_{\text{L}} = 1$ and $\beta_{\text{R}} = 0.5$, but equal linear density $\langle n \rangle_{\text{L}} = \langle n \rangle_{\text{R}} = 0.5$ and interaction strength $c = 1$, are merged. As shown in [45], the temperature difference alone causes a net flow from the hot side (right) towards the cold (left), as quasi-particles in the hot side generally travel faster.

The standard partitioning protocol features an abrupt transition between the two subsystems, however, this setup is not suitable for numerical calculation of correlations, as the initial inhomogeneity $\mathcal{D}_0$ of Eq. (20) is evaluated via finite difference. Instead, we employ a softened transition achieved via a steep hyperbolic tangent temperature profile. Meanwhile, the chemical potential was adjusted in order to maintain a constant linear density across the system.

In Fig. 4 we have plotted several quantities showing the propagation of density-density correlations, $\langle n(x,t)n(y,0) \rangle^{\text{Eul}}$. Subfigures 4(a) and 4(b) depict the spreading of the direct and indirect correlations for $y = 0$ respectively. Starting with the *direct* correlations, they appear very similar to the correlations in the homogeneous setup showcased earlier. This is somewhat expected, as the partitioning setup is (initially) piece-wise homogeneous with linear densities equal of the system in Subsec. 4.1. Upon closer inspection of Fig. 4(a), one might notice slightly higher correlations at the negative side (seen more clearly in Fig. 4(h)). This asymmetry reflects the net flow of quasi-particle from right to left, which is further exemplified in Fig. 4(g) showing the formation of the distinct, self-similar density profile as function of the ray, $\xi$. Moving on to the *indirect* correlations, we observe that the indirect correlations initially are antisymmetric around $x = 0$. As time passes, the indirect correlations become

more asymmetric due to the flow of particles.

The partitioning protocol is interesting from the viewpoint of correlations, since the inhomogeneities are very localized around $x = 0$, where the subsystems are joined. Therefore, it is interesting to vary $y$ such that it is not necessarily centered on the inhomogeneity. Subfigures 4(c-f) display the time-scaled correlation matrices, $t \langle n(x,t)n(y,0) \rangle^{\text{Eul}}$. Here, one clearly sees how the correlations start as delta functions, whereafter they propagate ballistically throughout the system. As the quasi-particles from the hot subsystem in general move faster, so do the correlations in that side propagate more rapidly. We see this in the correlation matrices, where one half of the correlations extend farther. Furthermore, towards the edges of the correlation matrices the correlations appear homogeneous, whereas around $x, y \approx 0$ a transition occurs. These three regions; the left side, the center, and the right side, are further explored in Subfig. 4(h), where $\langle n(x,1)n(y,0) \rangle^{\text{Eul}}$ is plotted for $y = -4, 0, 4$. The $y = 0$ profile we have already discussed: it is skewed toward the left due to the particle flow. Meanwhile, the remaining two profile are the result of placing the point $y$ within the homogeneous subsystems. The left subsystem exhibit a correlation profile very similar to the homogeneous system in Subsec. 4.1, as the two systems have identical temperatures. Again, the visible dip in correlations in the center of the profile is due to the high filling factor at lower rapidities present in the colder system. Conversely, the right profile exhibit no dip, as the subsystem is too hot to form any Fermi-sea-like quasi-particle distribution, whereby correlations can propagate freely even at low rapidity.

# 5 Conclusion

In this paper we have developed an iterative scheme (cf. Sec. 4 and Appendix A for further details) for finding the correlation propagator necessary for the calculation of exact, dynamical two-body correlation functions in the Eulerian limit (see Sec. 3 and Eqs. (18), (19) and (21) therein). We have applied our scheme to three different setups, whose transport properties have already been well-studied, namely an homogeneous thermal state in Subsec. 4.1, a bump release in Subsecs. 4.2 and 4.3, and a partitioning protocol in Subsec. 4.4. Furthermore, the universality of GHD enables our scheme to be applied to most integrable models. Thus, we have studied the spreading of correlations in the Lieb-Liniger model, the classical hard-rod model, and the relativistic sinh-Gordon model. In Subsec. 4.2, by comparing for the classical hard-rod model against the results obtained via Monte-Carlo simulations (the results are presented in Fig. 2), we have provided the first demonstration of the validity of the formulas derived in Ref. [1] for non stationary and inhomogeneous states. Crucially, we succeeded in explicitly confirming the subtle effect of indirect propagation of correlations – correlations due to the nonlinearity of GHD, and not directly interpreted as coming from the propagation of normal modes along their curved trajectories in the moving GHD fluid.

From this comparison we are able to observe the onset of the Eulerian limit at longer time-scales, while for short times deviations between the classical microscopic simulations and the Euler-scale generalized hydrodynamic predictions for correlation functions are observed. We expect that these discrepancies at smaller time-scales can be accounted by considering diffusive terms into Eq. (5). Although the effect of diffusive corrections on one-point functions is by now well understood [53–55], for dynamical two-point correlators in inhomogeneous and non-stationary states, instead, no analytical result is currently available. It would be interesting to investigate this point further in the future. Our results also enable us to analyze how statistical and dispersion properties of the various models affect the spreading of correlations, as shown in Subsec. 4.3.

Finally, we point out that our method not only enables future studies of correlation func-

tions, but it also allows one to extend the analysis of Refs. [77, 97, 98] for the calculation of the full counting statistics of the time integrated current of ballistically transported conserved quantities to the more complex and interesting case of inhomogeneous states in interacting integrable models. The authors plan to carry out this analysis in a future publication.

We have made the implementation of our numerical scheme public as a module of the iFluid framework for GHD calculations [79, 94].

## Acknowledgements

**Funding information**   F.M. acknowledges the support of the Doctoral Program CoQuS. This research was supported by the SFB 1225 'ISOQUANT' and grant number I3010-N27, financed by the Austrian Science Fund (FWF), and the Wiener Wissenschafts- und TechnologieFonds (WWTF) project No MA16-066 (SEQUEX). G.P. thanks the A. Della Riccia Foundation (Florence, Italy) – INFN for financial support and King's College (London, United Kingdom) for hospitality in the period when this work has been conceived and started.

## A   Numerical calculation of propagator

In order to solve Eqs. (19) and (21) we employ an iterative scheme. First, let the system be spatially discretized on a grid $x = \{x_0, x_1, \ldots, x_N\}$ where the first gridpoint, $x_0$, fulfills the same requirements as the lower limit of the integrals in Eqs. (19) and (21), namely that the filling function at values $x < x_0$ remain constant for the entire duration. Introducing the grid spacing $\bar{x}_n = x_{n+1} - x_n$ and the contraction $\left[ W_{(y,0) \to (x_n,t)} V^{\mathcal{O}'} \right](\lambda) \equiv W^{(n)}(\lambda)$, we can write Eq. (21) for the source term as follows

$$
\begin{aligned}
W^{(n)}(\lambda) = & -\sum_{m=0}^{n} \bar{x}_m \sum_{\gamma \in \lambda_*(x_m, t; y)} \frac{\rho_s(x_m, t; \gamma) \vartheta_0(y; \gamma) f(y, 0; \gamma)}{|\partial_\lambda \mathcal{U}(x_m, t; \gamma)|} T^{\mathrm{dr}}(x_m, t; \lambda, \gamma) V^{\mathcal{O}'}(\gamma) \\
& - \Theta(\mathcal{U}(x_n, t; \lambda) - y) \left( \rho_s(y, 0) f(y, 0) V^{\mathcal{O}'} \right)^{*\mathrm{dr}}(y, 0; \lambda) \\
= & \sum_{m=0}^{n} W_{(1)}^{(m)}(\lambda) + W_{(2)}^{(n)}(\lambda).
\end{aligned}
\tag{29}
$$

Separating the two terms, as done in the final line, highlights the iterative nature of the equation. Starting from $W^{(0)}(\lambda)$ at the very left side of the grid, one iterates over the spacial grid while updating the source term at each iteration.

Using the very same approach, the indirect propagator of Eq. (19) can be calculated. Let $\left[ \Delta_{(y,0) \to (x_n,t)} V^{\mathcal{O}'} \right](\lambda) \equiv \Delta^{(n)}(\lambda)$ and $\mathcal{D}_0(\mathcal{U}(x_n, t; \lambda) \equiv \mathcal{D}^{(n)}(\lambda)$ for a lighter notation. Then,

$$
\begin{aligned}
\Delta^{(n)}(\lambda) & = 2\pi \mathcal{D}^{(n)}(\lambda) \left( W^{(n)}(\lambda) + \sum_{m=0}^{n} \bar{x}_m \left( \rho_s(x_m, t) f(x_m, t) \Delta^{(m)} \right)^{*\mathrm{dr}}(\lambda) \right) \\
& = 2\pi \mathcal{D}^{(n)}(\lambda) \left( W^{(n)}(\lambda) + \sum_{m=0}^{n} W_{(3)}^{(m)}(\lambda) \right) \\
& = 2\pi \mathcal{D}^{(n)}(\lambda) \left( W^{(n)}(\lambda) + \sum_{m=0}^{n-1} W_{(3)}^{(m)}(\lambda) + \bar{x}_n \left( \rho_s(x_n, t) f(x_n, t) \Delta^{(n)} \right)^{*\mathrm{dr}}(\lambda) \right), \quad (30)
\end{aligned}
$$

where we with a slight abuse of notation have introduced the term $W_{(3)}^{(n)}$ as part of the indirect propagator to further emphasize the iterative nature of the equation.

Further examining Eqs. (29) and (30) provides the understanding necessary for choosing a suitable discretization. Starting with Eq. (29), its second term $W_{(2)}^{(n)}(\lambda)$ only contributes when $\mathcal{U}(x_n, t; \lambda) > y$, i.e. when any quasi-particles found at point $(x_n, t)$ originated to the right of $y$. By the definition of $x_0$, this term should vanish for $n \leq 0$. Meanwhile, the first term of (29) is dependent on all previous terms, however, only quasi-particles with rapidities within the root set $\lambda_*(x_m, t; y)$ contribute. In some non-relativistic models there is no upper limit to the quasi-particle velocity, like the Lieb-Liniger model where the group velocity is linearly proportional to the rapidity. Since the quasi-particle move according to a purely Eulerian equation with no inhomogeneous couplings, the overall rapidity distribution of the system is conserved throughout evolution. Thus, the grid must be chosen large enough such that no quasi-particle of the initial state has a large enough rapidity to reach $x_0$ within the time $t$. Finally, the name "source term" for $W^{(n)}(\lambda)$ becomes apparent when considering Eq. (30). Given the definition of $x_0$, the source term at the leftmost grid point must vanish, $W^{(0)}(\lambda) = 0$. Thus, the solution to Eq. (30) is likewise zero $\Delta^{(0)}(\lambda) = 0$. As we iterate over $x$, we eventually reach a point where quasi-particle starting at $y$ appear, whereby the source term becomes non-zero. This is turn results in a non-vanishing indirect propagator.

In order to solve (30) numerically, we have to discretize the rapidity as well. First, we rearrange the equation as follows

$$X^{(n)} \equiv 2\pi \mathcal{D}^{(n)} \left( W^{(n)} + \sum_{m=0}^{n-1} \mathcal{W}_{(3)}^{(m)} \right)$$

$$= \Delta^{(n)} - 2\pi \mathcal{D}^{(n)} \bar{x}_n \left( \rho_s(x_n, t) f(x_n, t) \Delta^{(n)} \right)^{*\mathrm{dr}}. \tag{31}$$

Next, we discretize the quantities following the notation of [79], where rapidity and type indices are lower and upper ones, respectively. In this notation the dressing operation of Eq. (4) reads $h^{\mathrm{dr}} = U^{-1} h$, where the matrix is $U_{ij}^{kl} = \delta_{ij} \delta^{kl} + w_j^l T_{ij}^{kl} \vartheta_j^l$. Since all the models treated in this paper only have a single quasi-particle type, we omit the type index for readability.

Hence, denoting $g^{(n)}(\lambda) \equiv \rho_s(x_n, t; \lambda) f(x_n, t; \lambda)$ the equation above can be written as

$$X_i^{(n)} = \Delta_i^{(n)} - 2\pi \mathcal{D}_i^{(n)} \bar{x}_n \left( \sum_j (U^{-1})_{ij} g_j^{(n)} \Delta_j^{(n)} - g_i^{(n)} \Delta_i^{(n)} \right)$$

$$= \sum_j \left( \delta_{ij} \left( 1 + 2\pi \mathcal{D}_j^{(n)} \bar{x}_n g_j^{(n)} \right) - 2\pi \mathcal{D}_j^{(n)} \bar{x}_n (U^{-1})_{ij} g_j^{(n)} \right) \Delta_j^{(n)}$$

$$\equiv \sum_j Y_{ij}^{(n)} \Delta_j^{(n)}, \tag{32}$$

where $X^{(n)}$ depends on the previous iterations. Thus, the calculation of the indirect propagator at the $n$'th grid point reduces to solving a linear matrix equation.

## B TBAs of studied models

The models used for studying the spreading of correlations are all very well known. We here briefly summarize their thermodynamic Bethe ansatz. A brilliant feature of GHD is the universality of the equations. Thus, each model needs only to provide a few specific quantities: the one-particle energy $\epsilon(\lambda)$, the one-particle momentum $p(\lambda)$, the differential two-body scattering phase $T(\lambda, \lambda')$, and the statistical factor $f(\lambda)$.

## B.1 Lieb-Liniger model

The Lieb-Linger model describes a one-dimensional Bose gas with contact interactions governed by the Hamiltonian [27, 28]

$$\hat{H} = \int_0^L dx \left\{ \partial_x \hat{\psi}^\dagger(x) \partial_x \hat{\psi}(x) + c\hat{\psi}^\dagger(x)\hat{\psi}^\dagger(x)\hat{\psi}(x)\hat{\psi}(x) - \mu \hat{\psi}^\dagger(x)\hat{\psi}(x) \right\}, \tag{33}$$

where $\hat{\psi}^\dagger(x), \hat{\psi}(x)$ are the bosonic fields, while $c$ is the interaction strength, $\mu$ is the chemical potential, and $\hbar = 2m = 1$. The TBA detailed here is only valid for repulsive interactions $c > 0$. Thus, the three main functions (single-particle energy, momentum and scattering) required for solving the GHD equations read

$$\epsilon(\lambda) = \lambda^2 - \mu \quad , \quad p(\lambda) = \lambda \quad , \quad T(\lambda, \lambda') = -\frac{1}{\pi}\frac{c}{c^2 + (\lambda - \lambda')^2} \, . \tag{34}$$

The quasi-particles in the Lieb-Liniger TBA are fermions, whereby their statistical factor reads

$$f(\lambda) = 1 - \vartheta(\lambda) \, . \tag{35}$$

## B.2 Relativistic sinh-Gordon model

The sinh-Gordon model is a relativistic field theory described by the Hamiltonian [96]

$$\hat{H} = \int dx \left\{ \frac{c^2}{2}\pi^2(x) + \frac{1}{2}[\partial_x \phi(x)]^2 + \frac{\beta^2 c^2}{g^2} : \cosh[g\phi(x)] : \right\}, \tag{36}$$

where

$$m^2 = \beta^2 \frac{\sin(\alpha \pi)}{\alpha \pi} \quad \text{and} \quad \alpha = \frac{cg^2}{8\pi + cg^2} \, . \tag{37}$$

In Eq. (36) $c$ denotes the velocity of light and we set $c = 1$ henceforth. Like the Lieb-Liniger model, the thermodynamic Bethe ansatz is determined by only a single root density. The TBA functions of the model read

$$\epsilon(\lambda) = m \cosh \lambda - \mu \quad , \quad p(\lambda) = m \sinh \lambda \quad , \quad T(\lambda, \lambda') = \frac{-1}{\pi}\frac{\sin(\pi \alpha) \cosh(\lambda - \lambda')}{\sin^2(\pi \alpha) + \sinh^2(\lambda - \lambda')}, \tag{38}$$

where we have added a chemical potential, $\mu$, to the energy function. Once again, the quasi-particles of the sinh-Gordon model are fermions, whereby

$$f(\lambda) = 1 - \vartheta(\lambda) \, . \tag{39}$$

Additionally, the expectation value of the vertex operator is given by [93, 99, 100]

$$\langle \Phi_k \rangle = \langle e^{kg\phi} \rangle = \prod_{j=0}^{k-1} H_j \, , \tag{40}$$

where

$$H_k = 1 + 4\sin(\pi\alpha(2k+1)) \int \frac{d\lambda}{2\pi} e^\lambda \vartheta(\lambda)\varepsilon_k^-(\lambda) \, , \tag{41}$$

and

$$\varepsilon_k^\pm(\lambda) = e^{\pm \lambda} + \int \frac{d\lambda'}{2\pi} 2\,\text{Im}\left(\frac{e^{2ki\pi\alpha}}{\sinh(\mp(\lambda - \lambda') - i\pi\alpha)}\right)\vartheta(\lambda')\varepsilon_k^\pm(\lambda') \, . \tag{42}$$

The one-particle form factor of the vertex operator used for calculating the correlations then reads

$$V^k(\lambda) = \frac{2}{\pi\rho_s(\lambda)}\sum_{j=0}^{k-1}\sin(\pi\alpha(2j+1))\varepsilon_j^+(\lambda)\varepsilon_j^-(\lambda)\prod_{\substack{i=0\\i\neq j}}^{k-1} H_l \, . \tag{43}$$

### B.3 Classical hard-rod model

The notion of integrability is not limited to quantum models but applies to some classical models as well. One of these models consists of hard rods of length $a$ on a one dimensional line [36,37,101]. Similar to models described above, a TBA description exists for the hard-rod model, where the relevant quantities are

$$\epsilon(\lambda) = \frac{\lambda^2}{2}, \quad p(\lambda) = \lambda \quad , \quad T(\lambda, \lambda') = \frac{a}{2\pi} . \tag{44}$$

For classical models, such as the hard-rod model, the statistical factor is merely

$$f(\lambda) = 1 . \tag{45}$$

The filling function for a thermal state can be easily written in terms of the source term $w^{(th)}(\lambda) = \beta \lambda^2/2$ as outlined in [77]

$$\vartheta^{(th)}(\lambda) = e^{-\varepsilon^{(th)}(\lambda)} = e^{-w^{(th)}(\lambda) - W(a\,d(\beta))} , \tag{46}$$

with the thermal pseudoenergy $\varepsilon^{(th)}(\lambda)$ being the solution of Eq. (1) with the source term $w^{(th)}(\lambda)$

$$\varepsilon^{(th)}(\lambda) = w^{(th)}(\lambda) + W(a\,d(\beta)) , \tag{47}$$

while $d(\beta)$ has been defined after (26) and $W(a\,d(\beta))$ is the Lambert-W function, which is defined as the solution of the equation

$$W = a\,d\,e^{-W} . \tag{48}$$

Similarly

$$\rho_s^{(th)}(\lambda) = \frac{1}{2\pi(1 + W(a\,d(\beta)))} , \tag{49}$$

whence, together with (3) the expression for the thermal root density $\rho_p^{(th)}(\lambda)$ in (26) immediately follows. From the latter the thermal linear density distribution of rods is constructed in the Monte-Carlo simulations as explained in Subsec. 4.2 of main text and in the following paragraph.

## C   Details of the Monte-Carlo simulations for the hard-rod gas

We present here additional details about the Monte-Carlo simulations presented in Subsec. 4.2. In our simulations we fix the initial point $-L$ ($L > 0$) whence rods are distributed in space and the number $N$ of particles. The initial position $L_M$ of the rightmost rod is therefore fluctuating for each different realization of the initial rods' configuration. The number $N$ of particles is chosen such that it is larger than the average number $\langle N \rangle$ of rods contained in the interval $[-L, L]$ (we take it symmetric for simplicity) where we want to compute the dynamics of the density $\langle n(x, t) \rangle$:

$$N > \langle N \rangle = \int_{-L}^{L} n(x, 0)\, dx . \tag{50}$$

The expression of the initial linear density $n(x, 0)$ used in the simulations is given in Eq. (27). As a consequence $L_M > L$. The simulations are performed in infinite volume, however, the initial rods' distribution is zero outside the interval $[-L, L_M]$ and there are two depletion zones that move inwards as time elapses in proximity of which the GHD solution does not hold anymore. Velocities are drawn from a Gaussian distribution with variance $1/\beta(x)$ according to

Eqs. (24) and (27). Notice that one can account for boosted thermal distributions by replacing $\lambda \to \lambda - \mu$ in Eq. (27). In all the simulations presented in the manuscript we have set for simplicity $\mu = 0$. The density and the two-point correlation function are computed by averaging over the number $M$ of independent sampled initial conditions

$$\langle n(x,t) \rangle_{\text{MC}} = \frac{1}{M} \sum_{i=1}^{M} \frac{N_i(x,t)}{l}, \tag{51}$$

$$\langle n(x,t)n(x,0) \rangle_{\text{MC}}^{\text{c}} = \frac{1}{M} \sum_{i=1}^{M} \frac{N_i(x,t)N_i(0,0)}{l^2} - \left( \frac{1}{M} \sum_{i=1}^{M} \frac{N_i(x,t)}{l} \right) \left( \frac{1}{M} \sum_{i=1}^{M} \frac{N_i(0,0)}{l} \right), \tag{52}$$

where $N_i(x,t)$ and $N_i(0,0)$ denote the number of rods at time $t$ in the interval $(x-l/2, x+l/2)$ and at time 0 in $(-l/2, l/2)$, respectively, for the $i = 1,2...M$ realization of the initial rods' positions and velocities.

The results obtained for a double thermal bump release on top of a constant thermal background for rod length $a = 0.1$ and inverse temperature profile $\beta(x)$ as per Eqs. (24) and (27) are further reported in Fig. 5 for completeness. The parameters are as follows: $x_0 = 300$, $\beta_{as} = 10$, $z = 120$, $\beta_{in} = 0.4$ and $a = 0.1$. The number of rods used in the Monte-Carlo simulations is $N = 270$, $L = 660$ and $l = 10$. For $t = 15$ and $t = 30$ we use $2 \cdot 10^6$ samples, while for $t = 70$ and 90, since the noise in the simulations increases, the sampling is enlarged to $7 \cdot 10^6$ and $8 \cdot 10^6$ samples, respectively.

Similarly to the cases analyzed in the main text, for the density dynamics $\langle n(x,t) \rangle$ an excellent agreement with the GHD results is obtained for all the times shown ($t = 15$, 30, 70 and 90). As far as correlation functions are concerned, for $t = 15$, 30 the discrepancy between the Euler-scale expression from Eqs. (18)-(21) and the Monte-Carlo results is evident, analogously to Fig. 2 in the main text. On the contrary, for longer times $t = 70$, 90 an excellent agreement is again attained. This aspect is witnessed by the relative distance $\sigma$ between the two methods, that for $t = 15$, 30 is significantly larger than the one for the values 70 and 90. For short time-scales, as a consequence, the Euler-scale formulas in Eqs. (18)-(21) are not directly applicable and diffusive corrections have to be included to correctly reproduce the correlation function dynamics, as commented also in the main text after Fig. 2. For $t = 70, 90$ the rods' density distribution gets smoothed by the dynamics, and the Euler-scale approximation in Eq. (7) becomes practically exact also for two-point correlation functions, since $\sigma$ is solely determined by the noise in the Monte-Carlo sampling.

In concluding this Appendix, we mention that to further explore the regime of parameters where Eqs. (18)-(21) are applicable we have also run simulations of the hard-rod dynamics for smaller values of $z$ compared to the ones used in Figs. 2 and 5. Specifically, we have considered the evolution from the two bumps initial state, akin to Fig. 5, but with $z = 60$, a number of rods $N = 140$, $x_0 = 150$, $L = 320$ and the other parameters ($\beta_{as}$, $\beta_{in}$ and $a$) having identical values to the ones used for Fig. 5. In this case, due to the smaller value of $z$ compared to Fig. 5, the rods' initial distribution is sharp and a small value of $l = 0.5$ has to be used for the averaging in space of the rods' positions. The number of samples $M$ has therefore to be greatly enlarged in order to reduce the noise from the statistical sampling. Despite this technical difficulty, which can be solved by performing longer simulations, we have observed an excellent agreement with the numerical simulations, similarly to Fig. 5. Note that for $z = 60$, and the aforementioned choice of the other parameters, the initial rods' density profile is made of two sharp bumps initially located at the positions $\pm x_0 = \pm 150$ with a small number of approximately 35 rods within each bump. It is therefore somehow unexpected that the Euler-scaling limit in Eq. (7) works so well even for such a small number of rods within the length scale of the initial inhomogeneity given by the width $z$ of the bumps. For

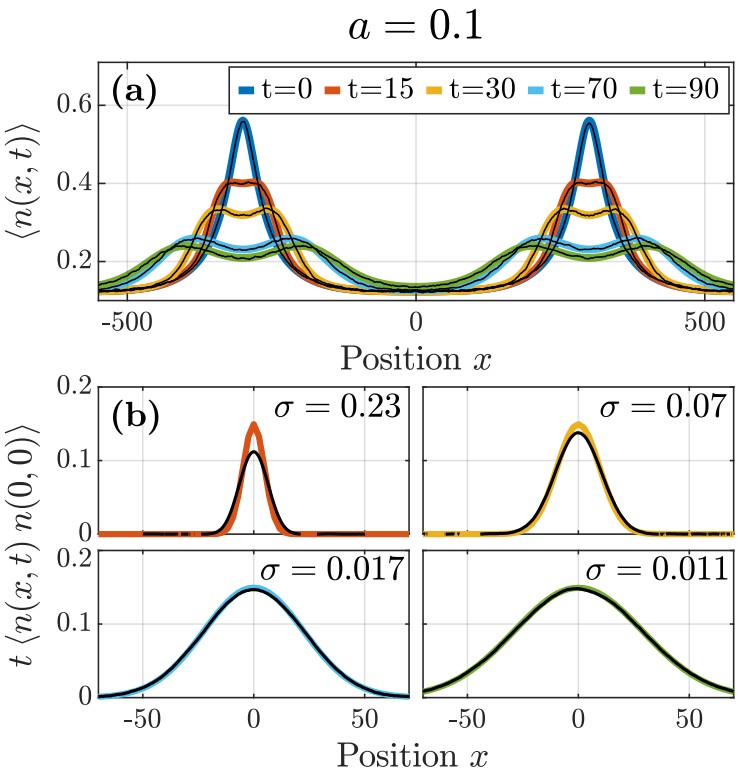

Figure 5: Release of a double density bump in the hard-rod model calculated using GHD (colored lines) and Monte-Carlo (black lines). Parameters of the Monte-Carlo simulations are specified in the text. **(a)** Comparison of the evolution of the linear density. **(b)** Comparison of two-point correlation $t \langle n(x,t)n(0,0) \rangle$ multiplied by time $t$ and the relative distance between the two approaches, $\sigma$.

the hard-rod model, as a matter of fact, it is generically difficult to get truly away from the Euler scaling limit, as shown in Ref. [36] where even for a very small number of 20 rods the results for the particle current and density (i.e., for one-point functions) obtained from the simulations of the hard-rod dynamics from the partitioning protocol initial state were shown to be very close to the Euler-scale formulas from GHD. The numerical analysis of the present manuscript, therefore, establishes that a similar behavior applies also to dynamical two-point functions, where deviations from the Euler-scale results in Eqs. (18)-(21) require an extremely sharp initial inhomogeneity, $z \sim 30$, with a number of approximately 15 rods contained in each bump.

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
