# Peer review of "Euler-scale dynamical correlations in integrable systems with fluid motion"

_SciPost Physics Core, doi:SciPost Phys. Core 3, 016 (2020)_

## Round 2 · Referee Report · Anonymous (Referee 1) · 2020-8-13

Report

The manuscript describes applications of a numerical method for calculation of dynamical two-point correlations to several integrable models. The paper can be interesting and useful for specialists in the field. However, I have several remarks. After proper revision, the paper can meet the acceptance criteria.

As the Euler scale is one of the most important concepts used in the paper, it should be defined and discussed in the introduction for the reader’s convenience.

The statement of the lack of thermalization in integrable systems may be misleading and requires more comprehensive discussion. Integrable systems do relax, but the final state is described by the generalized Gibbs ensemble rather than the Gibbs one. Moreover, there exist also incompletely-chaotic systems (see [PRL 106, 025303]), which relax to a state between the initial one and thermal equilibrium. The Authors also should include a discussion of the basic method of description of integrable systems - the coordinate Bethe ansatz, which can provide both correlation functions (see the review [Adv. At. Mol. Opt. Phys. vol. 55, 61]) and nonequilibrium dynamics (see, e.g., [PRL 119, 220401 (2017)]).

The relativistic sinh-Gordon model is considered only in Appendix B, although application of the method to this model is mentioned in the introduction. I don’t see a reason why the Authors decided don’t present the content of this appendix as a section in the main part.

P. 7: Probably, “which much be chosen” should be “ which must be chosen”

P. 10: A reference to the definition of the Lambert W function should be included. E.g., it can be dlmf.nist.gov if the Author’s definition is indeed the same.

Sec. 4.2: The Authors demonstrate only the examples where the approximation is working perfectly. It would be instructive to present also the examples when the approximation starts to fall, showing to the readers the applicability limits of the approximation.

Sec. 4.2: The direct and indirect correlations are undefined (these terms are also used previously, but I didn't find the definition). Probably, this means the contributions of the direct and indirect propagation, but the terms should be defined unambiguously.
  • validity: -
  • significance: -
  • originality: -
  • clarity: -
  • formatting: -
  • grammar: -

Author:  Frederik Skovbo Møller  on 2020-11-05  [id 1033]

(in reply to Report 1 on 2020-08-13)
Category:
answer to question

We thank the Referee for the careful reading of our manuscript. In the following, we addressher/his report.

As the Euler scale is one of the most important concepts used in the paper, it should be defined and discussed in the introduction for the reader’s convenience.

We thank the Referee for pointing out this point about the Euler scale. Indeed, the latter concept was not properly introduced in the manuscript despite its fundamental importance for the results presented. We have corrected this issue by first introducing the Euler scale in the introduction. In Sec. 2, after Eq. (6), we have explicitly stated in Eq. (7) how the Euler scaling limit is defined for one-point functions. At the beginning of Sec. 3, in Eq. (12), we have eventually defined the Euler scaling limit for two-point correlation functions.

The statement of the lack of thermalization in integrable systems may be misleading and requires more comprehensive discussion. Integrable systems do relax, but the final state is described by the generalized Gibbs ensemble rather than the Gibbs one. Moreover, there exist also incompletely-chaotic systems (see [PRL 106, 025303]), which relax to a state between the initial one and thermal equilibrium. The Authors also should include a discussion of the basic method of description of integrable systems - the coordinate Bethe ansatz, which can provide both correlation functions (see the review [Adv. At. Mol. Opt. Phys. vol. 55, 61]) and nonequilibrium dynamics (see, e.g., [PRL 119, 220401 (2017)]).

We agree with the Referee that the discussion in the original version of the manuscript about the “lack of thermalization” in integrable models was misleading and imprecise. We have revised the introduction by discussing more extensively how the concept of the generalized Gibbs ensemble (GGE) emerges in homogeneous quantum quenches in integrable models and we have emphasized that the relaxation to the GGE in integrable systems can be understood as a “generalized thermalization”, as pointed out, e.g., in Ref. [23], which we have added in the revised version of the paper. In addition, we have also included Refs.[15,16,19,20,22] regarding the GGE and homogeneous quantum quenches in integrable models. Moreover, as suggested by the Referee, we have included Ref. [20], where the coordinate Bethe ansatz is employed to study the dynamics of the one-dimensional interacting Bose gas after a quench of the interaction coupling, and Ref. [40], which contains applications of the coordinate Bethe ansatz to the calculation of correlation functions; see also Ref. [77], where the Bethe ansatz techniques are also used to study the non-equilibrium dynamics.

We have, however, not discussed in details the coordinate Bethe ansatz since our results are based on the thermodynamic Bethe ansatz (TBA) only, which has much wider applicability. For instance, there is no coordinate Bethe ansatz for the classical hard-rod model, which we study in this paper, while the TBA fully applies to this model, as explained in the cited literature. For the same reason, albeit very interesting, we have not included the discussion of incompletely chaotic systems, in PRL 106, 025303 suggested by the Referee, since our results rely solely on the GGE description via the TBA. We also mention that, as far as we are aware, it is not possible, in general, to obtain reliable Euler-scale correlation functions in moving nonzero-entropy fluids by coordinate Bethe ansatz techniques, such as those reproduced numerically here using the Monte Carlo technique in the hard-rod gas. These are obtained at scales of space and time which are not reachable by modern-day computers using the coordinate Bethe ansatz. This is one of the main reasons for the power of the GHD results based on TBA, which we here review, implement algorithmically and check numerically.

The relativistic sinh-Gordon model is considered only in Appendix B, although application of the method to this model is mentioned in the introduction. I don’t see a reason why the Authors decided don’t present the content of this appendix as a section in the main part.

We thank the Referee for pointing out this inconsistency in the way the results were presented in the manuscript. We have corrected it in the revised version of the manuscript by moving the Subsection “Comparing light cones of different models” from the Appendix to the main text(it is Subsection 4.3 of the revised manuscript).

P. 7: Probably, “which much be chosen” should be “ which must be chosen”

We thank the Referee for pointing out this misprint, which we have corrected in the revised version of the manuscript.

P. 10: A reference to the definition of the Lambert W function should be included. E.g., it can be dlmf.nist.gov if the Author’s definition is indeed the same.

We have added Ref. [91] right after Eq. (27) of the revised manuscript to make explicit that W(z) denotes the principal branch of the Lambert W function.

Sec. 4.2: The Authors demonstrate only the examples where the approximation is working perfectly. It would be instructive to present also the examples when the approximation starts to fall, showing to the readers the applicability limits of the approximation.

We thank the Referee for raising this question, which is fundamental to understand the results presented in the manuscript. We remark that the Euler-scale results in Eqs. (18)-(21) have to be understood as asymptotic expressions for the correlation functions valid in the limit of larges pace-time scales and variation length z of the inhomogeneous and non-stationary state,as commented after Eq. (7) and (12). Our numerical results in Fig. (2) and (5) confirm, indeed, this expectation since for short time-scales deviations from the hydrodynamic limit of the two-point correlation functions in Eqs. (18)-(21) are clearly visible and quantified by the errorσ.The plots for times t= 15 and 30 in Figs. (2) and (5) therefore already provide an example where the hydrodynamic approximation at the basis of Eqs. (18)-(21) does not apply. This is caused by the fact that for short times generalized thermalization in the fluid cells at the various space-time points (x,t) is not reached and therefore Eq. (7) approximate only poorly the actual dynamics of the correlations. For short times one should account for the large, but finite, variation length z of the initial inhomogeneous state and therefore one should look at the first correction beyond the Euler scaling limit in Eq. (7). In the revised manuscript, both around Eq. (28) and in the Appendix after Fig. (5), we have commented about this point and we have emphasized that the hydrodynamic approximation fails at short times for the description of two-point correlation functions. Furthermore, in the final part of the Appendix C, we have commented about the applicability of the hydrodynamic expressions in Eqs. (18)-(21) for small values of z. In particular, we have run simulations of the hard-rod dynamics from the two-bump initial state for smaller values of z= 60 compared to the ones considered in the main text and in the Appendix C (z= 120 and z= 200). Also in the case of z= 60 we have found an excellent agreement between Eqs. (18)-(21) and the Monte-Carlo simulations in a similar way as in Fig.(5). This shows that for the hard-rod gas the hydrodynamic approximation applies fairly well even for initial inhomogeneities varying on length scales z relatively small.

Sec. 4.2: The direct and indirect correlations are undefined (these terms are also used previously, but I didn't find the definition). Probably, this means the contributions of the direct and indirect propagation, but the terms should be defined unambiguously.

We have explicitly defined both the direct and indirect correlations after Eq. (18) in the revised version of the manuscript. In this way it is clear, as the Referee correctly suggests, that the direct correlations correspond to the contribution of the direct propagator, first line of the r.h.s.of Eq. (18), and the indirect correlations to the contribution of the indirect propagator, second line of the r.h.s. of Eq. (18). At various places in the manuscript, including in the abstract, introduction and in Section3 (in addition to the analysis which was already present in Section 4, e.g., Fig. 2), we have also emphasised the subtle effects due to indirect correlations – by contrast to those directly due to hydrodynamic modes co-moving with the fluid – in inhomogeneous, non-stationary, interacting fluids. Verifying their presence and the correctness of the indirect propagator, as confirmed by our numerical Monte Carlo results, was indeed one of the goals of this paper.

---

## Round 3 · Referee Report · Anonymous (Referee 1) · 2020-11-11

Report

The Authors have substantially improved the manuscript and, in my opinion, the revised version meets the acceptance criteria. However, I believe that the statement “integrable systems relax to stationary states which display non-thermal features, as a consequence of such conservation laws.” in Introduction is in the air with no mention of non-integrable systems which can either relax to the Gibbs state (the completely-chaotic systems with eigenstate thermalization) or to a non-thermal state (the incompletely-chaotic systems).
  • validity: -
  • significance: -
  • originality: -
  • clarity: -
  • formatting: -
  • grammar: -

Author:  Frederik Skovbo Møller  on 2020-11-17  [id 1047]

(in reply to Report 1 on 2020-11-11)

We thank the Referee for the second revision of the manuscript and for the supportive report. Indeed our exposition was not clear enough. We have revised the introduction of the manuscript by expanding the discussion about the relaxation of isolated many-body quantum systems at long times. We have specified more precisely the situation where the classification of stationary states is simplest: infinite-volume quenches from clustering states (which is the one we consider in this paper). There, one expects that the set of quasi-local conserved quantities fully specifies the stationary state; we do not think there are known exceptions to this. However, beyond this paradigm, indeed other possibilities occur,such as incompletely chaotic systems, as the Referee mentions. We have added a sentence pointing this out, including Ref. [24] about incompletely-chaotic systems, as suggested by the Referee. We have also included Refs. [21,22,23] about the eigenstate thermalization hypothesis (ETH) to make the discussion about the thermalization more complete. We emphasize, however, that we have discussed only some very fundamental points about thermalization of non-integrable systems, despite the rich literature available on this subject, since this topic is very marginally related to the original results presented in the manuscript.

---

## Round 3 · Author Response

Dear Editor,

Many thanks for your message and for forwarding us the report. We have answered all the questions of the referee report directly in the reply/comment below. We hope that the Referee will be satisfied by our revision and that the manuscript will be found ready for publication.

Best regards,
the Authors.

---

## Round 3 · List of Changes

We have added an additional sentence in the abstract concerning the subtlety of the indirect correlations.

The introduction has been expanded with a comprehensive discussion of the meaning of "thermalization" and "relaxation" in integrable systems. A discussion/definition of the Euler-scale has been added as well.

In the introduction, the subtle role of the indirect correlations is now further highlighted.

In the section 2 "Summary of GHD", we have moved the equations for the filling function and the dressing operation further up, whereby they now are eqs. (3) and (4).

In the section 2 "Summary of GHD", we have added a paragraph and eq. (7) detailing averages of local observables at the Euler-scale.

At the start of section 3 " Exact Euler-scale dynamical two-point correlations", we have added a definition of the Eulerian scaling limit for correlation functions.

In the section 3 " Exact Euler-scale dynamical two-point correlations", around eq. (18), we have made it more clear which terms correspond to the indirect correlations.

In section 4.2 "Bump release and Monte-Carlo comparison", we now further discuss the limits of the Euler-scale, and in which scenarios the GHD results do not apply.

We have added a new subsection 4.3 "Comparing light cones of different models" to the main text, which was previously found in the Appendix.

In Appendix C "Details of the Monte-Carlo simulations for the hard-rod gas", we have added an extension discussion of the limits of the Euler-scale results.

We have cited the references suggested by the referee in the manuscript.

---

## Round 5 · Author Response

Many thanks for your message and for forwarding us the report. Once again, we have answered all the questions of the referee report directly in the reply/comment below. We hope that the Referee will be satisfied by our revision and that the manuscript will be found ready for publication.
Best regards,
the Authors.

---

## Round 5 · List of Changes

Matthew Davis on 2020-11-21 [id 1052]
I have checked v5 - it appears to be the updated version. There was a problem at some stage with the SciPost page for v5 being broken - perhaps this was the origin of the issue.
Anonymous on 2020-11-19 [id 1048]
It is strange, but I don't see the last changes in the current version (v5 in arXiv). Probably, it is a technical mistake as the changes do present in the v4, which didn't appear in SciPost.
Anonymous on 2020-11-19 [id 1049]
(in reply to Anonymous Comment on 2020-11-19 [id 1048])Strange. For me, the current version (v5) shows up correctly (on two different devices) - both in arXiv and on the SciPost submission page. Note, between v4 and v5 we expanded the answer to the question posed regarding the relaxation of isolated many-body quantum systems. The changes should all be found on the first page of the introduction.

---

## Editorial Decision

published